# Holm Oak (*Quercus ilex* subsp. *ballota* (Desf.) Samp.) Bark Aqueous Ammonia Extract for the Control of Invasive Forest Pathogens

**DOI:** 10.3390/ijms231911882

**Published:** 2022-10-06

**Authors:** Eva Sánchez-Hernández, Joaquín Balduque-Gil, Juan J. Barriuso-Vargas, José Casanova-Gascón, Vicente González-García, José Antonio Cuchí-Oterino, Belén Lorenzo-Vidal, Jesús Martín-Gil, Pablo Martín-Ramos

**Affiliations:** 1Department of Agricultural and Forestry Engineering, ETSIIAA, University of Valladolid, Avenida de Madrid 44, 34004 Palencia, Spain; 2AgriFood Institute of Aragon (IA2), CITA-Universidad de Zaragoza, Avda. Montañana 930, 50059 Zaragoza, Spain; 3Instituto Universitario de Investigación en Ciencias Ambientales de Aragón, EPS, Universidad de Zaragoza, Carretera de Cuarte s/n, 22071 Huesca, Spain; 4Department of Agricultural, Forest and Environmental Systems, Agrifood Research and Technology Centre of Aragón, Instituto Agroalimentario de Aragón—IA2, CITA-Universidad de Zaragoza, Avda. Montañana 930, 50059 Zaragoza, Spain; 5Instituto Universitario de Investigación en Ingeniería de Aragón (I3A), EPS, University of Zaragoza, Carretera de Cuarte s/n, 22071 Huesca, Spain; 6Microbiology Service, Hospital Universitario Rio Hortega, Calle Dulzaina 2, 47012 Valladolid, Spain

**Keywords:** *Cryphonectria parasitica*, chestnut blight, *Fusarium circinatum*, in vitro tests, *Phytophthora cinnamomi*, pitch canker, root rot

## Abstract

Holm oak (*Quercus ilex* subsp. *ballota* (Desf.) Samp.) bark is a commonly used remedy to treat gastrointestinal disorders, throat and skin infections, hemorrhages, and dysentery. It has also been previously reported that its methanol extracts possess antibacterial activity, which can be related to the richness of *Quercus* spp. extracts in phenolic compounds, such as flavonoids and tannins. However, there is no information on the antifungal (including oomycete) properties of the bark from *Q. ilex* or its subspecies (*ilex* and *ballota*). In this work, we report the characterization of the aqueous ammonia extract of its bark by FTIR and GC-MS and the results of in vitro and ex situ inhibition tests against three phytopathogens. The main phytochemical components identified were inositols (19.5%), *trans*-squalene (13%), 4-butoxy-1-butanol (11.4%), gulopyranose (9.6%), lyxose (6.5%), 2,4-dimethyl-benzo[H]quinoline (5.1%), catechol (4.5%), and methoxyphenols (4.2%). The efficacy of the extract in controlling forest phytopathogens was tested in vitro against *Fusarium circinatum* (responsible for pitch canker of *Pinus* spp.), *Cryphonectria parasitica* (which causes chestnut blight), and *Phytophthora cinnamomi* (which causes ‘root and crown rot’ in a variety of hosts, including *Castanea*, conifers, *Eucalyptus*, *Fagus, Juglans, Quercus*, etc.), obtaining EC_90_ values of 322, 295, and 75 μg·mL^−1^, respectively, much lower than those attained for a commercial strobilurin fungicide (azoxystrobin). The extract was further tested ex situ against *P. cinnamomi* on artificially inoculated, excised stems of ‘Garnem’ almond rootstock, attaining complete protection at a dose of 782 μg·mL^−1^. The results suggest that holm oak bark extract may be a promising source of bioactive compounds against invasive forest pathogens, including the oomycete that is causing its decline, the so-called ‘seca’ in Spain.

## 1. Introduction

From ancient times to the present, the usage of barks has changed and expanded in response to various socioeconomic circumstances as well as scientific and technological advancements. Barks exhibit a great deal of diversity and are highly rich in chemical components, particularly in extractives such as sterols, terpenes, and numerous other phenolic compounds. These properties make barks useful as adhesives, formaldehyde scavengers, and antioxidants in medicine and pharmacy [1].

*Quercus ilex* L. (and its two subspecies *Quercus ilex* subsp. *ilex* L. and *Quercus ilex* subsp*. ballota* (Desf.) Samp.) is an evergreen tree that usually reaches 15 m in height. It has a broad canopy of ascending branches, and its relatively short trunk can sometimes exceed 2 m in diameter. As one of the most geographically widespread species in Portugal and Spain, *Q. ilex* and its subspecies have traditionally been considered important timber raw materials, apart from the use of acorn production as a staple food in the past. *Q. ilex* subsp*. ballota* is found in a wide variety of soils due to its seedling and root performance, stomatal responses, antioxidant systems, as well as its xylem plasticity, showing better drought resistance than *Q. ilex* subsp. *ilex* L. and deciduous *Q. faginea* Lam. species [2]. However, most studies have dealt with *Q. ilex* subsp. *ilex*, which is morphologically and genetically distinct from *Q. ilex* subsp. *ballota* and distributed differently. Likewise, the chemical components of holm oak bark have not been explored for the obtainment of high added-value products, such as biorationals for crops and forest species protection.

All organs of the plant contain tannins, and the tannin content can be quite high in the seeds. However, tannins have low toxicity and, because of their bitter taste and astringency, they are unlikely to be consumed in large quantities. Apart from tannins, the main chemical constituents of *Q. ilex* bark are suberin (ω-hydroxyacids), polysaccharides, lignin, and extractives [1,3]. Other constituents are catechins and phenolic acids (4-hydroxybenzoic, caffeic, coumaric, ferulic, and gentisic acids) [4]. As for the leaf composition of *Quercus* species, tannins, catechins, and phenolic acids (e.g., gallic acid, ellagic acid, protocatechuic acid, gentisic acid, chlorogenic acid, vanillic acid, syringic acid, epicatechin, naringenin, hesperetin, formononetin, naringin, kaempferol) have also been reported [5,6,7]. Concerning the main components of *Q. ilex* leaf oils, oleic acid, trans-2-hexanal, viridiflorol, and sabinene have been identified [8]. Information on the phenolics present in *Q. ilex* roots and acorn extracts, as well as on acorn oil, may be found in a recent review paper by Morales [9].

Some of the aforementioned products may be susceptible to valorization as ‘green agrochemicals’, with bactericidal, fungicidal, antiviral, insecticidal, acaricidal, or nematicidal activities. Berahou et al. [10], in a study on the antibacterial properties of holm oak bark, concluded that the phytochemicals present in ethyl acetate and *n-butanol* extracts and the aqueous layer were effective against seven reference bacterial strains: *Escherichia coli* ATCC 11775, *Pseudomonas aeruginosa* Schroeter 1872 ATCC 27853, *Staphylococcus aureus* Rosenbach 1884 BCCM 21055, *Bacillus subtilis* (Ehrenberg 1835) ATCC 6051, *Klebsiella pneumoniae* Schroeter 1886 ATCC 13883, *Salmonella typhimurium* (Le Minor et al. 1982, Le Minor and Popoff 1987) ATCC 43971, *Vibrio cholerae* Pacini 1854 ATCC 14033, *Proteus mirabilis* Hauser 1885 HITM 20, *S. epidermidis* (Winslow and Winslow 1908) Evans 1916 HITM 60, *Streptococcus pyogenes* Rosenbach 1884 HITM 100, and *S. agalactiae* Lehmann and Neumann 1896 HITM 80, with MIC values ranging from 128 to 512 μg·mL^−1^. 

In the work presented here, apart from the characterization of the phytochemicals contained in the oak bark extract by gas chromatography-mass spectrometry (GC–MS), the results of the study of the antifungal and anti-oomycete activity of an aqueous ammonia extract for the control of invasive forest pathogens (including the oomycete that is causing the decline of oak in the Iberian Peninsula) are presented. In particular, the efficacy of this extract has been assayed against *Fusarium circinatum* Nirenberg and O’Donnell (teleomorph *Gibberella circinata*) (EU quarantine pathogen that causes pitch canker of *Pinus* spp. and *Pseudotsuga*
*menziesii* (Mirb.) Franco), *Cryphonectria*
*parasitica* (Murril) M.E. Barr (responsible for chestnut blight), and *Phytophthora cinnamomi* Rands (which causes ‘root and crown rot’ in a wide range of hosts, mainly belonging to the genera *Castanea*, *Eucalyptus, Fagus, Juglans, Quercus*, etc.).

## 2. Results

### 2.1. Phytochemicals Identified by GC–MS

The main phytochemicals identified in the aqueous ammonia extract from the bark of *Q. ilex* subsp. *ballota* (Appendix A, Table 1) were: inositols (*myo*-inositol, *allo*-inositol and 1-deoxy-inositol) (19.5%), *trans*-squalene (13%), 4-butoxy-1-butanol (11.4%), gulopyranose (9.6%), lyxose (6.5%), 2,4-dimethyl-benzo[H]quinoline (5.1%), catechol (4.5%), 1-pentanol (4.5%), methoxyphenols (4.2%), and 2-hydroxy-2-cyclopenten-1-one (2.7%). Their chemical structures are presented in Figure 1.

### 2.2. In Vitro Antimicrobial Activity

Mycelial growth inhibition tests against the three phytopathogens were conducted for the bark extract and its two main constituents. The increase in the assayed dose resulted in statistically significant differences and in full inhibition in all cases (Figure 2). The inhibitory activity of *Q. ilex* subsp. *ballota* bark extract (Figure 2a and Appendix A) was significantly higher against *P. cinnamomi* than against *F. circinatum* and *C. parasitica* (for which total inhibition was achieved at similar concentrations, as shown by the EC_90_ effective concentration values summarized in Table 2). Upon testing the two main constituents, namely *myo*-inositol (Figure 2b and Appendix A) and *trans*-squalene (Figure 2c and Appendix A), it could be observed that the latter showed much higher efficacy than the former, attaining full inhibition of *F. circinatum, C. parasitica,* and *P. cinnamomi* at concentrations as low as 250, 187.5, and 85.9 μg·mL^−1^, respectively (vs. 1000, 750 and 375 μg·mL^−1^, respectively, for *myo*-inositol). 

The activity of the holm oak extract was compared with that of a commercial systemic, broad-spectrum fungicide frequently used in forestry and agriculture to control numerous plant diseases, viz. Azoxystrobin. This popular synthetic pesticide of the strobilurin family [11] inhibits pathogens’ ability to respire their mitochondria and impairs a number of biological and biochemical functions of living cells (by preventing electron transfer, stopping the synthesis of adenosine triphosphate, and disrupting the flow of energy) [12]. At the manufacturer’s recommended dose (62.5 mg·mL^−1^), 68%, 81.3%, and 92% inhibition was attained against *F. circinatum, C. parasitica*, and *P. cinnamomi*, respectively (corresponding to radial growth values of 24, 14, and 6 mm, respectively). 

### 2.3. Protection of Excised Stems against P. cinnamomi

*Ex situ* tests conducted on almond rootstock ‘Garnem’ excised stems were conducted to assess the efficacy of the treatment against the phytopathogen for which the best in vitro results have been attained, viz. *P. cinnamomi*. At the lowest assayed dose, i.e., the MIC value obtained in the in vitro tests (78.2 μg·mL^−1^), no protection was observed, with canker lengths similar to those of the untreated stems. At five times the MIC dose (391 μg·mL^−1^), large cankers were also registered, with no significant differences vs. the control. It was necessary to increase the dosage up to 10 times the MIC (782 μg·mL^−1^) to obtain full protection of the excised stems, as shown in Figure 3. At this concentration, no signs of fungal colonization were observed in any of the replicates, neither in the outer bark nor in the cambium tissues. 

To fulfill Koch’s postulates, samples from both cankers of inoculated ‘Garnem’ stems and colonized PDA plates were taken apart and mounted on a microscope slide with 3% KOH as mounting media and morphologically inspected to confirm the identity of the microorganism responsible for the lesions. These microscopical observations confirmed the presence of somatic and reproductive structures compatible with those of *P. cinnamomi*.

## 3. Discussion

### 3.1. On the Phytochemical Composition

Differences in the phytochemical profiles between those reported in the literature [1,3,4,5,6,7,8] and the one described in this study should be mainly ascribed to differences in extraction media and methodology, as well as in the method of acquisition [7], although slight differences between subspecies cannot be ruled out. Whereas the results of this study were obtained for an aqueous ammonia extract and applying GC–MS, the results by Meziti et al. [4] and Hadidi et al. [6] corresponded to hydromethanolic extracts characterized by high-performance liquid chromatography (HPLC) and high-performance liquid chromatography with a diode array detector (HPLC–DAD), respectively; the thorough study conducted by Sousa et al. [1] used dichloromethane as the extraction medium and GC–MS; Karioti et al. [5] chose diethylether for the extraction and used HPLC–DAD–MS; and Dallali et al. [8], also in diethylether medium, determined the constituents of the essential oils by gas chromatography with flame-ionization detection (GC−FID). However, the main constituents identified in the present study have been reported for other plant extracts in the literature: for instance, inositol was previously identified in *Abrus precatorius* L. seeds [13], *Cocos nucifera* L. [14], *Crinum latifolium* L. leaves [15], and *Rhizophora apiculata* Blume roots [16]. *Myo*-inositol has been found in the latex of *C. bonplandianum* L. [17]. Squalene is a component of *Cuscuta reflexa* Roxb.*, A. precatorius, Abutilon indicum* L.*, Acalypha indica* L.*, Ammannia baccifera* L. [18]*, C. maxima* Duchesne [19]*, Jasminum grandiflorum* L. [20], and *Leucas aspera* (Willd.) Link [21]. Concerning the dialkyl ether 4-butoxy-1-butanol, it has been identified in *Apium graveolens* L. leaves by Nagella et al. [22].

### 3.2. Antimicrobial Activity Comparison

Regarding the antimicrobial activity of *Q. ilex* extracts, few studies are available in the literature (Appendix A). As for leaf extracts, Boy et al. [23] evaluated the efficacy of *Q. ilex* leaf extracts obtained by ultrasonication in 90% (*v*/*v*) ethanol-water against yeasts responsible for food spoilage, finding a high inhibition capacity against *Candida boidinii* C. ramirez*, Priceomyces carsonii* (Phaff et Knapp) Suzuki et Kurtzman*, Kregervanrija fluxuum*, (Phaff and E.P. Knapp) Kurtzman and *Zygosacharomyces bailii* (Lindner) Guillierm at a concentration of 2000 µg·mL^−1^, attributed to the high content of phenolic compounds; Sánchez-Gutiérrez et al. [24] demonstrated the antibacterial activity of an aqueous extract of *Q. ilex* leaves obtained by a microwave-assisted extraction procedure against foodborne Gram- and Gram+ bacteria with inhibition values in the 1000 to 5000 µg·mL^−1^ range; and Güllüce et al. [25] tested in vitro a methanol extract of *Q. ilex* leaves against a wide range of human and plant-associated microorganisms, including *Pseudomonas syringae* Van Hall 1904, with a MIC of 250 µg·mL^−1^. Berahou et al. [10] tested different solvents (ethyl acetate, *n*-butanol, or water) to prepare *Q. ilex* bark extracts, and assayed them against 11 reference bacterial strains, finding inhibition values ranging from 128 to 512 μg·mL^−1^; Merghache et al. [26] showed the inhibitory capacity of *Q. ilex* wood ashes against *Candida albicans* at a 5% concentration; and Bakour et al. [27] studied the antibacterial effect of the ethanolic extract of pollen from six botanicals, including *Q. ilex*, finding that it was the most efficient against multidrug-resistant bacteria. 

To date, and to the best of the authors’ knowledge, no data on the antifungal/anti-oomycete action of *Q. ilex* bark extracts have been reported. However, results on the antifungal activity of other species of the genus *Quercus* are available. For example, *Q. suber* L. bark acetone extract showed strong antifungal activity against *Trichophyton verrucosum* E. Bodin and *T. mentagrophytes* C.P. Robin) R. Blanch., with MIC values of 20 µg·mL^−1^ [28]. Antifungal activity against *Aspergillus flavus* Link*, A. ochraceus* Willhem*, A. niger* Tiegh.*,*
*C. albicans* (C.P. Robin) Berkhout*, Penicillum feniculosum* Thom, and *P. ochrochloron* Biourge has also been observed for methanolic extracts of *Q. acutissima* Carruth*., Q. macrocarpa* Michx., and *Q. robur* L. bark, with MIC values ranging from 160 to 2000 µg·mL^−1^ [29]. Dania et al. [30] found that *Q. phillyreoides* A. Gray bark aqueous extract, at a concentration of 3%, resulted in strong inhibition of mycelial growth (>77%) in in vitro tests against *Rhizoctonia solani* J.G. Kühn, *Lasiodiplodia theobromae* (Pat.) Griffon and Maubl.*, A. niger, Sclerotium rolfsii* Sacc.*, P. oxalicum* Currie and Thom, and *F. oxysporum* Schltdl. Inhibitions in the range of 60–80.5% against *C. glabrata* (H.W. Anderson) S.A. Mey. and Yarrow*, C. albicans, A. niger, A. flavus, F. solani*, and *Microsporum canis* E. Bodin ex Guég. have been reported for some compounds present in the methanolic extract of *Q. incana* W. Bartram bark at a concentration of 5000 µg·mL^−1^ [31]. Hence, the results obtained for the aqueous ammonia extract under study (with EC_90_ values between 75 and 322 μg·mL^−1^) are among the lowest reported for *Quercus* spp. and may be regarded as very promising, especially in the case of *P. cinnamomi*.

### 3.3. Comparison of Efficacy vs. Other Natural Compounds

Table 3 summarizes a literature search on the inhibitory values of natural compounds evaluated against *F. circinatum*, *C. parasitica,* and *P. cinnamomi* with those obtained in this investigation. The MIC values given below should be taken with caution, given that the susceptibility profile depends on the isolates as well as on the testing methods used, and provided that units substantially differ (e.g., when essential oils are used as biofumigants, MICs are expressed in μg∙mL^−1^ air, and the MICs of compounds tested by the agar/liquid dilution methods are expressed in µg·mL^−1^). 

In the case of *F. circinatum*, the activity of the aqueous ammonia extract of *Q. ilex* subsp. *ballota* would be higher than those of cinnamon, fennel, clove [32], thyme, rose geranium, and lemon grass essential oils [33] (assuming that many essential oils have a density somewhere in the vicinity of 0.9 g·mL^−1^). As noted above, lower MIC values reported by Lee et al. [34] and Lee et al. [35] may not be compared, as in those studies the essential oils were used as biofumigants. The same applies to the MIC values reported by Lee et al. [34], Lee et al. [35], Lukovic et al. [36], and Kim et al. [37] against *C. parasitica.*

As for *P. cinnamomi*, holm oak bark extract activity would be lower than that of ethanolic plant extracts of *Larrea tridentata* (Sessé and Moc. ex DC.) Coville and *Flourensia cernua* DC. (Hojasé, Hojasén) (EC_90_ values of 11.19 and 23.61 μg∙mL^−1^, respectively), and the lanolin extract of *Agave lechuguilla* Torr. (MIC_90_ = 58.3 μg∙mL^−1^) [38], and higher than those of, for example, the most effective essential oils reported by Giamperi et al. [39] (those of *Origanum vulgare* L. and *Thymus vulgaris* L., with MICs ≥ 200 μg∙mL^−1^) and of the essential oil of aerial parts of *Beilschmiedia miersii* (Gay) Kosterm. (MIC = 300 μg∙mL^−1^) [40].

### 3.4. Comparison with a Conventional Fungicide

Concerning the chemical fungicide chosen as a reference, viz. azoxystrobin, it is one of the world’s biggest selling fungicides [43]. Data on its efficacy against the three phytopathogens under study is scarce in the literature. Benalcázar Villalba [44] reported EC_50_ values in the 49.8–263.8 µg·mL^−1^ range against *Fusarium* spp. associated with the death of *Pinus radiata* D. Don. and *P. patula* Schiede ex Schltdl. and Cham. seedlings in the nursery, and González-Varela and González [45] found that it was unable to stop *C. parasitica* growth at doses in the 1–16 µg·mL^−1^ range. 

Based on the tests conducted in this study at the recommended dose (62.5 mg·mL^−1^), at which full inhibition was not reached, it may be inferred that the activity of the holm oak bark extract would be much higher (with EC_90_ values ranging from 75 to 322 μg·mL^−1^).

### 3.5. Mode of Action

To provide a tentative explanation of the observed antifungal and anti-oomycete activity, a description of each of the main extract constituents and their previously reported activities is first presented. Inositol is a group of nine stereoisomers, but the name is usually used to describe the most common type of inositol, *my*o-inositol (*cis*-1,2,3,5-*trans*-4,6-cyclohexanehexol), considered a pseudovitamin. It has been reported to be an efficient adjuvant to antibiotic drugs, increasing their antimicrobial activity [46]. In a recent study, Ratiu et al. [47] examined 40 different plant species that contained variable quantities of *myo-*inositol. The fruits of blueberries, lettuce, and cinnamon had the greatest concentrations (0.96, 1.07, and 1.21 mg·g^−1^ dry plant material, respectively). Supraene (*trans*-squalene) is a linear triterpene synthesized in plants, animals, bacteria, and fungi as a precursor for the synthesis of secondary metabolites such as sterols, hormones, or vitamins. It is known to have active oxygen-scavenging activities, preventing oxidative damage. Concerning its fungicidal action, also reported for other supraene-rich natural products (summarized in Appendix A [48,49,50,51]), it is known that the intracellular accumulation of squalene disrupts fungal cell membranes, possibly by forming squalene vesicles that weaken fungal cells by extracting essential membrane lipid components [52]. In fact, the mechanism of action of terbinafine and other antifungal drugs is based on the inhibition of squalene peroxidase, resulting in the aforementioned squalene accumulation [53]. The dialkyl ether 4-butoxy-1-butanol has been identified in benzene/alcohol extractives of *Artocarpus lingnanensis* Merr. [54]. As regards 2,4-dimethyl-benzo[H]quinolone, it should exhibit antifungal activity, like other benzoquinoline derivatives [55]. Methoxyphenols as 2-methoxyphenol, 2,6-dimethoxyphenol, and 3,4,5-trimethoxy-phenol are among the phenolic compounds produced by alkaline destruction of poplar wood bark and exhibit antioxidant properties [56] and anti*-*quorum sensing effects [57,58]. For the minority component 2-hydroxy-2-cyclopent-2-en-1-one, Naika and Pavani [59] advocated the existence of antibacterial activity.

Based on the above-discussed activities and taking into consideration the EC_90_ values for the two main constituents of the aqueous ammonia extract (between 321 and 710 μg·mL^−1^ for *myo*-inositol and ranging from 88 to 174 μg·mL^−1^ for squalene), the antimicrobial activity of the extract may be mainly ascribed to the *trans*-squalene content, whereas inositol would have a weaker activity and would act as an adjuvant of squalene. Nonetheless, contributions from other minor constituents such as 2,4-dimethyl-benzo[H]quinoline and 3,4,5-trimethoxyphenol cannot be ruled out (and their possible synergies may account for the higher efficacy of the bark extract against *P. cinnamomi* in comparison to that of pure squalene). Further research on the activity of these phytochemicals and their combinations is needed to elucidate the mechanism of action of the bark extract.

## 4. Material and Methods

### 4.1. Reagents

Ammonium hydroxide, 50% *v*/*v* aq. soln. (CAS No. 1336-21-6), was purchased from Alfa Aesar (Ward Hill, MA, USA). Acetic acid (purum, 80% in H_2_O; CAS No. 64-19-7); *myo*-inositol (≥99%; CAS No. 87-89-8), squalene (analytical standard; CAS No. 111-02-4), and potato dextrose agar (PDA) were supplied by Sigma Aldrich Química S.A. (Madrid, Spain); Alkir^®^ fungicide co-adjuvant (ROPF No. 19454) was purchased from De Sangosse Ibérica (Valencia, Spain).

The commercial fungicide used for comparison purposes, viz. Ortiva^®^ (azoxystrobin 25%; reg. no. 22000; Syngenta, Basel, Switzerland) was kindly provided by the Plant Health and Certification Service (CSCV) of Gobierno de Aragón (Zaragoza, Spain). This fungicide was selected due to its low risk of resistance, widespread use, favorable toxicological and environmental profile, and significant pathogen control ability.

### 4.2. Phytopathogen Isolates

*Fusarium circinatum*, *Cryphonectria parasitica*, and *Phytophthora cinnamomi* isolates were supplied as PDA subcultures by the Calabazanos Forest Health Center (Villamuriel de Cerrato, Palencia, Spain).

### 4.3. Plant Material

The holm oak bark was torn from an old specimen from the holm oak forest of Nisano (Huesca, Spain). The sample was shade-dried and pulverized to a fine powder in a mechanical grinder. 

### 4.4. Preparation of the Extract

An aqueous ammonia solution was chosen to dissolve the bioactive compounds of interest contained in the bark of holm oak. The bark extract was prepared according to the procedure described in [60]. Briefly stated, a probe-type ultrasonicator (model UIP1000hdT; 1000 W, 20 kHz; Hielscher Ultrasonics, Teltow, Germany) was used to sonicate the bark sample for 10 min in pulse mode with a 2 min break after every 2.5 min of sonication, and the sample was then left to settle for 24 h. Acetic acid was then used to change the pH to neutral. After 15 min of centrifuging the solution at 9000 rpm, the supernatant was filtered using Whatman No. 1 paper.

### 4.5. Characterization of the Extract

The aqueous ammonia extract of oak bark was analyzed by gas chromatography-mass spectrometry (GC–MS) at the Research Support Services (STI) at the University of Alicante (Alicante, Spain), using a gas chromatograph model 7890A coupled to a quadrupole mass spectrometer model 5975C (both from Agilent Technologies, Santa Clara, CA, USA). The operating conditions were: injector temperature = 280 °C, splitless mode; injection volume = 1 µL; initial temperature = 60 °C, 2 min, followed by a ramp of 10 °C·min^−1^ to a final temperature of 300 °C, 15 min. The chromatographic column used for the separation of the compounds was an Agilent Technologies HP-5MS UI of 30 m in length, 0.250 mm diameter, and 0.25 µm film. The mass spectrometer conditions were: mass spectrometer electron impact source temperature = 230 °C and quadrupole temperature = 150 °C; ionization energy = 70 eV. Test mixture 2 for apolar capillary columns according to Grob (Supelco 86501) and PFTBA tuning standards were used for calibration, purchased from Sigma Aldrich Química S.A. (Madrid, Spain). Comparison of mass spectra and retention times with those of reference compounds and computer matching with the databases of the National Institute of Standards and Techniques (NIST11) and the monograph by Adams [61] were used for compound identification.

### 4.6. In Vitro Antimicrobial Activity Assessment

The antimicrobial activity of the treatment was assessed according to EUCAST standard antifungal susceptibility testing techniques [62], using the agar dilution method. Aliquots of stock solution were mixed into a pouring PDA medium to obtain concentrations ranging from 62.5 to 1500 μg·mL^−1^ (albeit lower concentrations of 15.62 and 31.25 μg·mL^−1^ also had to be evaluated for the most efficient treatments to obtain reliable PROBIT fittings). Plugs (Ø = 5 mm) from the margins of one-week-old PDA cultures of *F. circinatum*, *C. parasitica*, and *P. cinnamomi* were transferred to plates integrating the above treatment concentrations (three plates per treatment and concentration, with two duplicates). Plates were incubated at 25 °C in the dark for one week. PDA media without any modification was employed as a control. Growth inhibition was estimated according to the formula ((*d_c_* − *d_t_*)/*d_c_)* × 100, where *d*_c_ and *d*_t_ indicate the mean diameters of the control and treated colonies, respectively. The effective concentrations (EC_50_ and EC_90_) were determined using PROBIT analysis in IBM SPSS Statistics v.25 software (IBM; Armonk, NY, USA).

### 4.7. Protection Tests on Artificially Inoculated Excised Stems

Given the restrictions that apply to in vivo assays involving *P. cinnamomi*, the efficacy of the treatment was tested by artificial inoculation of excised stems in controlled laboratory conditions. Inoculation was performed according to the procedure proposed by Matheron [63], with modifications. Using a grafting knife, young stems of healthy ‘Garnem’ (*Prunus amygdalus* × *P. persica*) rootstock with a 1.5 cm diameter were cut into 10 cm-long sections. The excised stem pieces were immediately wrapped in moistened sterile absorbent paper, and the wounds produced were painted with Mastix^®^.

In the laboratory, the freshly excised stem segments were first immersed in ethanol for 1 min, then immersed in a NaClO solution for 8 min, and then thoroughly rinsed with distilled water [64]. Some of the stem segments (*n* = 15) were soaked for 1 h in distilled water as a control, and the remaining stem segments were soaked for 1 h in aqueous solutions to which an appropriate amount of the bark extract had been added to obtain MIC, MIC × 5, and MIC × 10 concentrations (*n* = 15 segments/concentration). Alkir^®^ co-adjuvant (1% *v*/*v*) was added to all solutions (including the control) to facilitate bark penetration of the treatment.

The stem pieces were allowed to dry, placed on an agar Petri dish, and subsequently (Ø = 5 mm) from the margin of one-week-old PDA cultures of *P. cinnamomi* on the center of the outer surface of the bark. After inoculation, the stem segments were incubated in a humid chamber for 4 days at 24 °C, 95–98% RH. 

The efficacy of the treatments was assessed by visual inspection of the presence of rotting at the inoculation sites, confirmed under the microscope, both on the outer bark and on the inner bark after careful removal with a scalpel to reveal the cambium. Then, the oomycete was re-isolated and morphologically identified from the lesions to fulfill Koch’s postulates.

### 4.8. Statistical Analysis

The results of the postharvest protection study were statistically analyzed in IBM SPSS Statistics v.25 software by analysis of variance (ANOVA), followed by a post hoc comparison of means by Tukey’s test (because the requirements of homogeneity and homoscedasticity were met, according to the Shapiro–Wilk and Levene tests).

## 5. Conclusions

In vitro tests on the antifungal and anti-oomycete activities of the aqueous ammonia extract of holm oak bark aimed at controlling important forest phytopathogens such as *F. circinatum*, *C. parasitica*, and *P. cinnamomi,* have resulted in EC_90_ values of 322, 295, and 75 μg·mL^−1^, respectively. Such activity, which may be ascribed to *trans*-squalene content (13%), was shown to be much higher than that of azoxystrobin, tested for comparison purposes, and higher than most of those reported in the literature for other natural products against these forest pathogens. Although a higher dose (782 μg·mL^−1^) was needed to attain full inhibition in further tests conducted on excised almond ‘Garnem’ stems artificially infected with *P. cinnamomi*, the results suggest that *Q. ilex* subsp. *ballota* bark extract may be a promising source of bioactive compounds against phytopathogens.

## Figures and Tables

**Figure 1 ijms-23-11882-f001:**
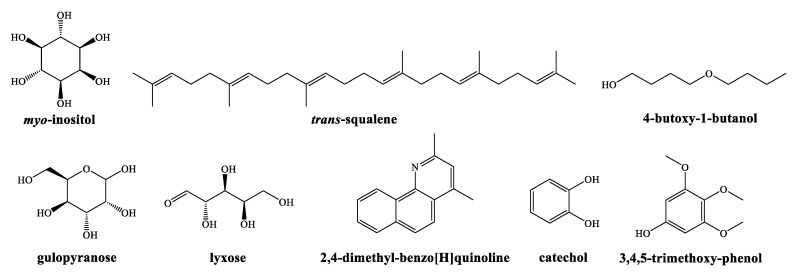
Main phytochemicals identified in the aqueous ammonia extract of *Q. ilex* subsp*. ballota* bark.

**Figure 2 ijms-23-11882-f002:**
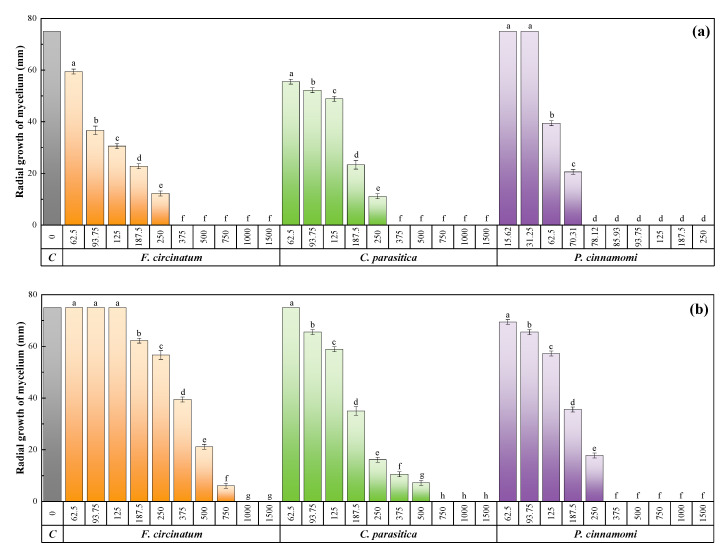
Mycelial growth inhibition attained with (**a**) *Q. ilex* subsp*. ballota* bark extract, (**b**) *myo*-inositol, and (**c**) *trans*-squalene against *F. circinatum*, *C. parasitica,* and *P. cinnamomi* at concentrations ranging from 62.5 to 1500 μg·mL^−1^ (or from 15.6 to 250 μg·mL^−1^ for *Q. ilex* subsp*. ballota* bark extract and *trans*-squalene against *P. cinnamomi*). The same letters above concentrations indicate that they are not significantly different at *p* < 0.05. Error bars represent standard deviations.

**Figure 3 ijms-23-11882-f003:**
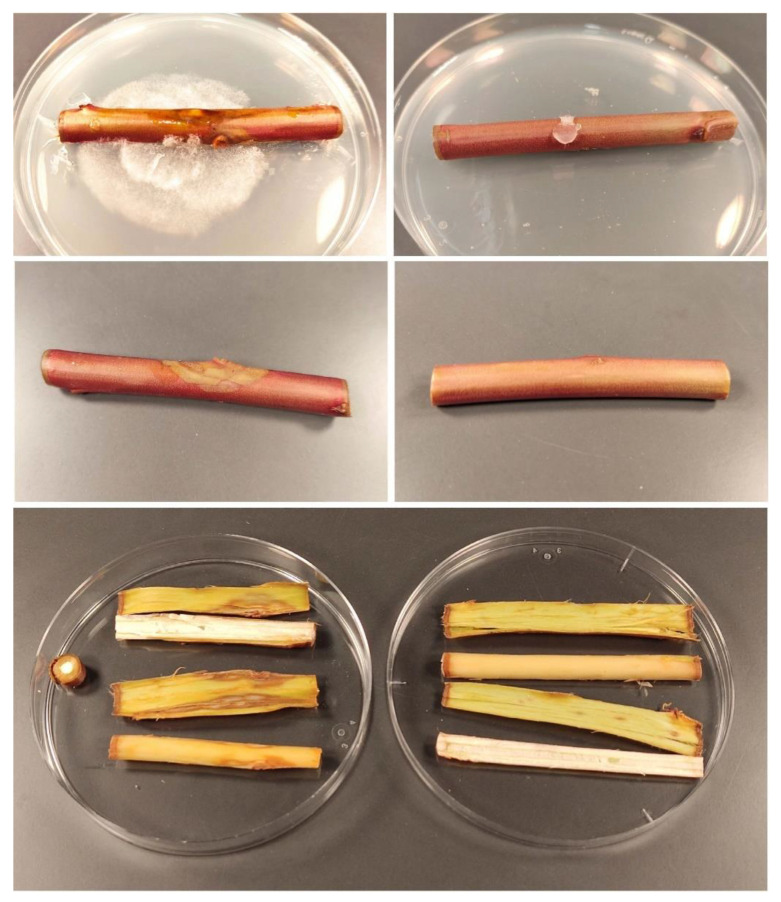
‘Garnem’ stem segments artificially inoculated with *P. cinnamomi* after 4 days of incubation: (**left**) untreated samples; (**right**) samples treated with holm oak bark extract at a 782 μg·mL^−1^ (MIC×10) dose.

**Table 1 ijms-23-11882-t001:** Most representative phytochemicals identified in *Q.*
*ilex* subsp*. ballota* bark aqueous ammonia extract by GC–MS.

Peak	RT (min)	Area (%)	Assignment
1	4.3897	4.3045	oxime-, methoxy-phenyl-_
2	4.6983	4.5463	1-pentanol
3	4.7695	2.7271	2-cyclopent-2-en-1-one, 2-hydroxy-
4	5.7607	2.2880	succindialdehyde
5	5.8379	3.3537	2-hydroxy-*γ*-butyrolactone
6	7.2861	1.2101	2-methoxy-phenol
7	7.3573	1.4969	pentanal
8	8.9064	4.4657	catechol
9	9.0489	2.3118	1H-tetrazole, 5-(trifluoromethyl)-
10	9.8620	1.1263	pyridine, 4-nitro-, 1-oxide
11	10.3843	0.5025	1H-imidazole-4-methanol, 5-methyl-
12	11.0491	1.2235	2,6-dimethoxy-phenol
13	11.6664	1.1952	3-octyne
14	12.2183	11.4443	1-butanol, 4-butoxy-
15	12.5804	1.0696	2-trifluoroacetoxytridecane
16	14.2245	1.7480	3,4,5-trimethoxy-phenol
17	14.8833	6.9425	*allo*-inositol
18	14.9961	3.8720	inositol, 1-deoxy-
19	15.0258	0.9823	inositol, 1-deoxy-
20	15.0910	3.6660	d-lyxose
21	15.1563	2.8651	l-lyxose
22	15.2691	6.2190	d-gulopyranose
23	15.3225	3.3908	d-gulopyranose
24	15.3463	7.6969	*myo*-inositol
25	17.9103	1.2863	n-nexadecanoic acid
26	25.0920	12.9624	supraene (or *trans*-squalene)
27	26.6352	0.9232	benzo[H]quinoline, 2,4-dimethyl-
28	28.8194	1.3713	benzo[H]quinoline, 2,4-dimethyl-
29	29.5494	2.8086	benzo[H]quinoline, 2,4-dimethyl-

**Table 2 ijms-23-11882-t002:** EC_50_ and EC_90_ effective concentrations (in μg·mL^−1^) of *Q. ilex* subsp*. ballota* bark extract and its main constituents against the pathogens under study.

Product	Effective Concentration	*F. circinatum*	*C. parasitica*	*P. cinnamomi*
*Q. ilex* subsp*. ballota* bark extract	EC_50_	92.1	142.3	63.4
	EC_90_	322.4	294.9	75.2
*myo*-inositol	EC_50_	375.9	171.8	174.9
	EC_90_	710.2	472.6	321.5
*trans*-squalene	EC_50_	106.4	59.0	38.2
	EC_90_	173.6	135.2	87.8

**Table 3 ijms-23-11882-t003:** Inhibition values reported in the literature for other bioactive natural products against the three pathogens under study.

Pathogen	Source	Natural Product	Inhibitory Value	Ref.
*F. circinatum*	Aqueous ammonia bark extract (1:1)	*Quercus ilex* subsp*. ballota*	MIC = 375 µg·mL^−1^	This work
	Commercial EOs	*Artemisa arborescens* EO	n.a.	[34]
		*Anthemis nobilis* EO	n.a.	
		*Coriandrum sativum* EO	MIC > 28 µg·mL^−1^ air	
		*Cyperus scariosus* EO	MIC > 28 µg·mL^−1^ air	
		*Commiphora myrrha* EO	MIC > 28 µg·mL^−1^ air	
		*Pastinaca sativa* EO	MIC > 28 µg·mL^−1^ air	
		*Pogostemon patchouli* EO	MIC > 28 µg·mL^−1^ air	
		*Miroxylon balsamum* EO	MIC > 28 µg·mL^−1^ air	
		*Salvia stenophylla* EO	n.a.	
		*Santalum album* EO	n.a.	
		*Santolina chamaecyparissus* EO	n.a.	
		*Nardostachys sinensis* EO	n.a.	
		*Liquidambar orientalis* EO	MIC > 28 µg·mL^−1^ air	
		*Valeriana wallichii* EO	MIC > 28 µg·mL^−1^ air	
		*Lippia javanica* EO	n.a	
		*Leptospermum scoparium* EO	MIC > 28 µg·mL^−1^ air	
		*Juniperus mexicana* EO	n.a	
		*Daucus carota* EO	MIC > 28 µg·mL^−1^ air	
		*Calitis intratropica* EO	MIC > 28 µg·mL^−1^ air	
	Commercial EOs	*Eucalyptus citriodora* EO	MIC > 28 µg·mL^−1^ air	[35]
		*Melaleuca quinquenervia* EO	MIC > 28 µg·mL^−1^ air	
		*L. petersonii* EO	MIC > 28 µg·mL^−1^ air	
	Foliage, wood, and bark	*Cryptomeria japonica* EO	n.a.	[41]
	Commercial EOs	*Syzygium aromaticum* EO	MIC = 400–500 µL·L^−1^	[33]
		*Cymbopogon citratus* EO	MIC = 400–700 µL·L^−1^	
		*Thymus vulgaris* EO	MIC = 500 µL·L^−1^	
		*Pelargonium graveolens* EO	MIC = 900–1000 µL·L^−1^	
	n.e.	*Cinnamomum verum* EO	MIC = 10% *v*/*v*	[32]
		*Foeniculum vulgare* EO	MIC = 50% *v*/*v*	
		*S. aromaticum* EO	MIC = 15% *v*/*v*	
*C. parasitica*	Aqueous ammonia bark extract (1:1)	*Q. ilex* subsp*. ballota*	MIC = 375 µg·mL^−1^	This work
	Commercial EOs	*A. arborescens* EO	MIC > 28 µg·mL^−1^ air	[34]
		*A. nobilis* EO	n.a	
		*C. sativum* EO	n.a	
		*C. scariosus* EO	MIC > 28 µg·mL^−1^ air	
		*C. myrrha* EO	n.a	
		*P. sativa* EO	MIC > 28 µg·mL^−1^ air	
		*P. patchouli* EO	MIC > 28 µg·mL^−1^ air	
		*M. balsamum* EO	MIC > 28 µg·mL^−1^ air	
		*S. stenophylla* EO	MIC > 28 µg·mL^−1^ air	
		*S. album* EO	MIC > 28 µg·mL^−1^ air	
		*S. chamaecyparissus* EO	MIC > 28 µg·mL^−1^ air	
		*N. sinensis* EO	MIC > 28 µg·mL^−1^ air	
		*L. orientalis* EO	MIC > 28 µg·mL^−1^ air	
		*V. wallichii* EO	MIC > 28 µg·mL^−1^ air	
		*L. javanica* EO	MIC > 28 µg·mL^−1^ air	
		*L. scoparium* EO	n.a	
		*J. mexicana* EO	n.a	
		*D. carota* EO	n.a	
		*C. intratropica* EO	n.a	
	Commercial EOs	*E. citriodora* EO	MIC > 28 µg·mL^−1^ air	[35]
		*M. quinquenervia* EO	MIC > 28 µg·mL^−1^ air	
		*L. petersonii* EO	MIC > 28 µg·mL^−1^ air	
	Foliage, wood, and bark	*C. japonica* EO	n.a.	[41]
	n.e.	*Illicum verum* EO	MIC > 0.32 µg·mL^−1^ air	[36]
		*J. oxycedrus* EO	MIC = 0.08–0.16 µg·mL^−1^ air	
		*E. globulus* EO	MIC = 0.08–0-16 µg·mL^−1^ air	
		*Lavandula angustifolia* EO	MIC > 0.32 µg·mL^−1^ air	
		*Citrus limon* EO	MIC > 0.32 µg·mL^−1^ air	
		*C. flexuosus* EO	MIC > 0.32 µg·mL^−1^ air	
		*Mentha piperita* EO	MIC = 0.02 µg·mL^−1^ air	
		*Origanum vulgare* EO	MIC = 0.16–0.32 µg·mL^−1^ air	
		*Pinus nigra* EO	MIC = 0.02 µg·mL^−1^ air	
		*P. pinaster* EO	MIC = 0.16–0.32 µg·mL^−1^ air	
		*P. silvestris* EO	MIC = 0.32 µg·mL^−1^ air	
		*Rosmarinus officinalis* EO	MIC ≥ 0.32 µg·mL^−1^ air	
		*S. officinalis* EO	MIC = 0.04 µg·mL^−1^ air	
		*Abies alba* EO	MIC = 0.02 µg·mL^−1^ air	
		*Gaultheria procumbens* EO	MIC > 0.32 µg·mL^−1^ air	
	Commercial EOs	*Pimenta racemosa* EO	MIC > 28 µg·mL^−1^ air	[37]
		*J. oxycedrus* EO	MIC > 28 µg·mL^−1^ air	
		*C. nardus* EO	MIC > 28 µg·mL^−1^ air	
		*P. graveolens* EO	MIC > 28 µg·mL^−1^ air	
		*Cuminum cyminum* EO	MIC > 28 µg·mL^−1^ air	
		*Myristica fragrans* EO	MIC > 28 µg·mL^−1^ air	
		*C. martini* EO	MIC > 28 µg·mL^−1^ air	
		*M. pulegium* EO	MIC > 28 µg·mL^−1^ air	
		*M. spicata* EO	MIC > 28 µg·mL^−1^ air	
		*T. vulgaris* EO	MIC = 14 µg·mL^−1^ air	
*P. cinnamomi*	Aqueous ammonia bark extract (1:1)	*Q. ilex* subsp*. ballota*	MIC = 78.12 µg·mL^−1^	This work
	Water, ethanol (70%), lanolin (10%), or cocoa butter (10%)	*L. tridentata* PE	MIC_90_ = 11.2−7213 µg·mL^−1^	[38]
		*F. cernua* PE	MIC_90_ = 23.6−619 µg·mL^−1^	
		*A. lechuguilla PE*	MIC_90_ = 58.5−327 µg·mL^−1^	
		*Opuntia ficus-indica* PE	MIC_90_ = 3595−409, 181 µg·mL^−1^	
		*L. graveolens* PE	MIC_90_ = 4825−n.a. µg·mL^−1^	
		*Carya illinoensis* PE	n.a.	
		*Yucca filifera PE*	n.a.	
	n.e.	*S. officinalis* EO	MIC > 1600 µg·mL^−1^	[39]
		*S. rosmarinus* EO	MIC > 1600 µg·mL^−1^	
		*O. vulgare* EO	MIC > 200 µg·mL^−1^	
		*Laurus nobilis* EO	MIC > 1600 µg·mL^−1^	
		*C. sativum* EO	MIC = 800 µg·mL^−1^	
		*T. vulgaris* EO	MIC = 200 µg·mL^−1^	
		*M. piperita* EO	MIC = 800 µg·mL^−1^	
		*L. intermedia* EO	MIC = 1600 µg·mL^−1^	
	Aerial parts	*Beilschmiedia miersii* EO	MIC = 300 µg·mL^−1^	[40]
	Leaf methanol extract (1:5)	*Arbutus unedo* PE	MIC = 5990 µg·mL^−1^	[42]

PE = plant extract; EO = essential oil; n.a. = no activity; n.e. = not specified.

## Data Availability

The data presented in this study are available on request from the corresponding author. The data are not publicly available due to their relevance to an ongoing Ph.D. thesis.

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
