# Peer review of "Holm Oak (Quercus ilex subsp. ballota (Desf.) Samp.) Bark Aqueous Ammonia Extract for the Control of Invasive Forest Pathogens"

_ijms, 2022, doi:10.3390/ijms231911882_

Round 1

Reviewer 1 Report

The article written clearly. There is no doubt about the experiments carried out. The data obtained are substantiated both by the authors and the cited literature. There is a small note: the effect of the Q. ilex subsp. ballota bark extract on fungi is being investigated in comparison with myo-inositol, and trans-squalene, which are only part of the bark extract. For the purity of the data obtained, it is necessary to conduct tests with a mixture of myo-inositol, and trans-squalene. Since the effect of trans-squalene on F. circinatum, C. parasitica and P. cinnamomi was more effective than bark extract.

Author Response

The article is written clearly. There is no doubt about the experiments carried out. The data obtained are substantiated both by the authors and the cited literature.

Response: We thank the Reviewer for his/her positive feedback.

Q1. There is a small note: the effect of the Q. ilex subsp. ballota bark extract on fungi is being investigated in comparison with myo-inositol, and trans-squalene, which are only part of the bark extract. For the purity of the data obtained, it is necessary to conduct tests with a mixture of myo-inositol, and trans-squalene. Since the effect of trans-squalene on F. circinatum, C. parasitica and P. cinnamomi was more effective than bark extract.

Response: The experiment suggested by the Reviewer is interesting and we agree that it would provide additional information about the mode of action of the extract, but we feel that the exploration of such possible synergistic behavior between trans-squalene and myo-inositol falls outside the scope of this study, designed to focus specifically on the bark extract: the rationale behind the testing of the two main constituents of the extract was to gain some insight into the mechanism of action of the extract, but -as discussed in section 3.5- that first-approximation also involved excluding the in vitro testing of other phytoconstituents present in the extract in lower amounts that may also have antimicrobial activity (and with which synergies may also occur). We have now acknowledged this and suggested it as a topic for further research in section 3.5 of the revised manuscript: “Nonetheless, contributions from other minor constituents such as 2,4-dimethyl-benzo[H]quinoline and 3,4,5-trimethoxyphenol cannot be ruled out (and their possible synergies may account for the higher efficacy of the bark extract against P. cinnamomi in comparison to that of pure squalene). Further research on the activity of these phytochemicals and their combinations is needed to elucidate the mechanism of action of the bark extract.”

Reviewer 2 Report

The Authors submitted an interesting paper. They studied the possibility to use natural substances (flavonoids and tannins) extracted from bark of Quercus ilex, as antifungal agents. They tested these substances against three phytopathogens which cause severe disease in several plants. All the sections of the manuscript are clear and detailed, Materials and Methods mainly.The explanation and discussion of the results are good and complete. The references are adequate. The paper could give important and useful information to the researchers which work in this study field and could give good ideas to those who approach this type of study,

Only two suggestions: please clarify the use of the letters on the bars of figure 2,  and check that you have written in full the name of each microorganism and fungus at the first citation.

In my opinion the paper could deserve the pubblication

Author Response

The Authors submitted an interesting paper. They studied the possibility to use natural substances (flavonoids and tannins) extracted from the bark of Quercus ilex, as antifungal agents. They tested these substances against three phytopathogens which cause severe disease in several plants. All the sections of the manuscript are clear and detailed, Materials and Methods mainly. The explanation and discussion of the results are good and complete. The references are adequate. The paper could give important and useful information to the researchers which work in this study field and could give good ideas to those who approach this type of study.

Response: We thank the Reviewer for his/her positive feedback.

Only two suggestions:

Q1. please clarify the use of the letters on the bars in figure 2

Response: The same letters above concentrations indicate that they are not significantly different at p < 0.05, as stated in Figure 2 caption. We have now added this to the main text for clarity: “Mycelial growth inhibition tests against the three phytopathogens were conducted for the bark extract and its two main constituents. The increase in the assayed dose resulted in statistically significant differences and full inhibition in all cases (Figure 1). The inhibitory activity of Q. ilex subsp. ballota […]”

Q2. check that you have written in full the name of each microorganism and fungus at the first citation.

Response: We thank the reviewer for bringing this point to our attention. We have ensured that the name of each microorganism and fungus has been written in full (including the authority) at the first citation in the text.

In my opinion, the paper could deserve the publication